# Antibody and cytokine levels in visceral leishmaniasis patients with varied parasitemia before, during, and after treatment in patients admitted to Arba Minch General Hospital, southern Ethiopia

Dagimawie Tadesse[1]☯*, Alemseged Abdissa[2,3‡], Mekidim Mekonnen[3‡], Tariku Belay[3‡], Asrat Hailu[4]☯*

1 Department of Medical Laboratory Science, College of Medicine and Health Science, Arba Minch University, Arba Minch, Ethiopia, 2 Armauer Hansen Research Institute, Addis Ababa, Ethiopia, 3 School of Medical Laboratory Sciences, Faculty of Health Sciences, Jimma University, Jimma, Ethiopia, 4 Departments of Microbiology, Immunology, and Parasitology, Faculty of Medicine, Addis Ababa University, Addis Ababa, Ethiopia

☯ These authors contributed equally to this work.
‡ AA, MM and TB also contributed equally to this work.
* dagimawie@yahoo.com (DT); hailu_a2004@yahoo.com (AH)

**Data Availability Statement:** All relevant data are within the manuscript and its files.

## Abstract

### Background

Visceral leishmaniasis is a disease caused by disseminated Leishmania donovani infection which affects almost half a million people annually. Most of the patients are reported from the Indian sub-continent, Eastern Africa and Brazil. In this study, we aimed to determine the levels of antibodies and cytokines in visceral leishmaniasis patients and to examine associations of parasitemia with the clinical states of patients. A prospective study was carried out, enrolling a total of 48 active VL patients who were evaluated before, during different time points and, three months after treatment. Serum cytokine concentrations, antibody levels, parasitemia, laboratory (hematologic and biochemical) measurements, and clinical parameters were assessed.

### Results

Counts of WBC and platelets, and measurements of hemoglobin (Hb) increased during treatment (P ≤ 0.05). Elevated levels of circulating IL-10, IFN-γ, and TGF-β1 were measured before treatment. The observed increase in serum IL-10 remarkably declined within 7 days after the start of treatment. Anti-leishmanial antibody index (AI) was high in all VL patients irrespective of spleen aspirate parasite grade before treatment and at different times during treatment. However, a significant (P ≤ 0.05) decrease of AI was observed 120 days post-treatment. IL-2 serum levels were below the detection limit at all sampling points.

**Funding:** The authors received no specific funding for this work.

**Competing interests:** The authors have declared that no competing interests exist.

## Conclusions

The present results suggest that IL-10, IFN-γ, and TGF-β1 can be used as markers of active visceral leishmaniasis. In addition, measuring circulating cytokines concentrations, particularly IL-10, in combination with other clinical evaluations, could be used as criteria for the cure. The observation that a high serum concentration of IFN-gamma at baseline was associated with low parasitemia deserves further investigations.

## Author summary

Visceral leishmaniasis (also known as kala-azar) is a neglected tropical disease that occurs in widely dispersed areas of the world, including Ethiopia. Parasites in the *Leishmania donovani* complex are responsible for causing visceral leishmaniasis. The condition is difficult to diagnose and treat. We investigated how the immune response generated during follow-up treatment periods of active VL before, during, and post-treatment was influenced by the presence of different cytokines. It is important to identify possible immunological biomarkers that could be correlated with patients' clinical and parasitological presentation as well as the response patterns to treatment in VL patients of southwestern Ethiopia.

## Introduction

Visceral Leishmaniasis is a life-threatening parasitic infection that is endemic in 62 countries worldwide, including the Mediterranean region [1]. More than 90% of global VL cases occur in six countries: India, Bangladesh, Sudan, South Sudan, Ethiopia, and Brazil [2]. VL is the second largest cause of parasitic death after malaria and is responsible for almost 500,000 new cases and more than 50,000 deaths per year [2,3]. Anthroponotic VL is found in *L. donovani* transmission areas typically in the Indian subcontinent. The Eastern Africa region was second in disease burden next to the Indian subcontinent, primarily in Sudan, South Sudan, and Ethiopia, with an estimated 30,000–57,000 cases/year [2–5]. Presently, the region is considered to have the highest-burden. In Ethiopia, with an estimated 4,500 to 5,000 new cases occur every year, the population at risk is more than 3.2 million [6,7]. The challenges of VL control stem from the lack of appropriate tools for early diagnosis and treatment.

VL symptoms often persist for several weeks to months before patients either seek medical care, or die from bacterial co-infections (Tuberculosis and bacterial pneumonia), massive bleeding or severe anemia [5,8]. Early case-finding and treatment are considered an essential component of VL control [9]. The development of simple and rapid diagnostics to guide treatment and for epidemiological surveillance is the first step to achieve the goal of VL elimination. In patients being treated, a test of cure (ToC) would help assess response to treatment. Unfortunately, to date, there are no such reliable tests other than parasitology. Parasitological examination of bone marrow or lymph node examination is sub-optimal for use as a ToC as a negative parasitology result does not rule out sub-microscopic parasitemia or leishmaniasis infection in primary cases [10].

Concerning prognostic biomarkers, it is a worthful effort to test if concentrations of selected cytokines in serum could be used to monitor treatment outcomes. The acute phase of VL is associated with elevated expression of interferon-gamma (IFN-γ) mRNA in lesional

tissue, such as the spleen and bone marrow as well as increased circulating levels of multiple pro-inflammatory cytokines and chemokines, including interleukin-12 (IL-12), IFN-γ and TNF-α [11,12]. Higher plasmatic levels IFN-γ were also associated with clinical manifestations observed in active VL [13]–a phenomenon that depicts Th1 response countering replication of parasites [12]. The increasing plasma concentrations of IFN-γ and other pro-inflammatory cytokines are often accompanied by increases in concentrations of regulatory cytokines such as IL-10 and TGF-β [14]. Like IL-10 and IL-4, TGF-β has been implicated in the pathology of both experimental and human leishmanial disease advancement [15]. The production of TGF-β by infected macrophages is associated with inhibition of IFN-γ production, suppression of macrophage activation, and advancement of the disease processes.

The outcome of the typical symptomatic VL is influenced by the immune response developed by the host wherein the systemic infection, with the spread of the parasites to the spleen, liver, lymph nodes, bone marrow, and other organs are accompanied by a high titer of circulating antibodies, marked up-regulation of IL-4 and IL-10 [16–18]–and unresponsiveness to Type-1 T-cell mediated immunity. Sub-sets of regulatory Th cells share the important task of controlling over-exuberant immune responses utilizing IL-10 production [17–20]. In human studies, increased plasma levels of IL-10 have been reported during active VL [20–22]–and cure of disease is associated with a fall in IL-10 mRNA levels [23]. High antibody titers and immune-complex formation may contribute to the elevated IL-10 levels observed in VL patients and the progressive decline in the immune status of VL patients [24,25]. The above shreds of evidence suggest that IL-10 plays a key role in the immunological pathways leading to the systemic spread of Leishmania parasites in human VL.

For effective control of VL and in monitoring treatment responses, reliable and rapid diagnostic and prognostic tools are needed. Therefore, we launched a clinical laboratory study to identify cytokine biomarkers that could be correlated with antibody levels, parasitemia, and clinical presentation as well as response to treatment in VL patients of southwestern Ethiopia.

## Materials and methods

### Ethics statement

Ethical clearance was obtained from the Institutional Review Board of Jimma University, Institute of Health, School of Medical Laboratory Sciences (Ref. No. RPGC/2095/015). Written informed consent was sought from each individual before involvement in the study. Consent for the inclusion of young children (under 18 years) was obtained from their parents or guardians and assent from children of age 12–17. Examination of patients' and specimen collections was performed as recommended by the national guidelines and procedures. Consent was taken from all study participants in the storage of samples for further analysis and study. All patient-related data were handled with strict confidentiality.

### Study area

This study was conducted at Leishmaniasis Research and Treatment Center (LRTC), Arba Minch General Hospital (AMGH) in Gamo Gofa Zonal Administration, Southern Nations, Nationalities' and Peoples' Region (SNNPR), southern Ethiopia. Arba Minch is the administrative and commercial town of Gamo Zone, located 505 km from Addis Ababa. The total area of Arba Minch town was estimated at 1095 hectares and it lies at an altitude of 1300 meters above sea level; its average temperature is 29˚C and the average annual rainfall is 900 mm. VL patients present themselves to the treatment center or are referred from other health facilities.

## Study design and population

A descriptive cross-sectional study was conducted among admitted VL patients who were treated and followed at different time points between 02/September/2015 to 01/August/2016. The study population comprised of all VL cases. Based on composite diagnostic criteria [CoDC] the definitive diagnosis was made by either of the following: (1) presence of definitive clinical signs and symptoms as well as positive antibody test with rk39 rapid test combined with a favorable response to treatment, and/or (2) parasitological confirmation by smear and/or culture.

## Study participants and sampling

A total of 48 patients admitted in LRTC for VL treatment who fulfilled the inclusion criteria were enrolled consecutively. The mean treatment provided was the one recommended by the Ethiopian MoH guideline combining (20mg/kg b.wt) of Sodium Stibogluconate (SSG) and (15mg/kg b.wt) of Paromomycin (PM) for 17 days. However, due to shortcut of PM during the study period 11 patients from the 48 ones received only SGG for the same dose with an extension of duration until 30 days. For control values of cytokines and antibodies analyses, plasma of 20 healthy blood donors was procured from the Ethiopian National Blood Bank.

## Laboratory investigations

Before the study enrollment, presumptive VL diagnosis was made based on the WHO clinical case definition: prolonged fever (at least 2 weeks), weight loss, and splenomegaly in a patient from an endemic area or with travel history (S1 Table). In patients who meet the case definition, the rK39 dipstick test (InBios International, Inc. Seattle, WA, USA) was performed as part of standard of care "triage" to rule in suspicion of VL and to assess enrollment eligibility. Following a preliminary positive screening test by rk39, clinical and hematological investigations and direct microscopic examination of splenic or bone marrow aspiration were done systematically. When splenic aspiration was contraindicated, only bone marrow aspiration was performed. In addition, a test of cure was not possible to perform in the majority of patients who showed clinical cure and regression of the enlarged spleen, in which case only bone marrow punctures could be performed. The detection of Leishmania donovani bodies in in tissue aspirates (spleen and, bone marrow) of patients with symptoms was a definitive confirmation of VL. Routine laboratory test results were also captured, including testing for HIV [either by Voluntary Counseling and Testing or by Provider-Initiated Counseling and Testing] and blood chemistry.

After the informed consent, 7.5 ml of whole blood was collected from each study participants in a plain and sodium-EDTA vacutainer (Becton, Dickinson, UK) tubes. Serum/plasma was separated for quantitative assessment of both antibody levels and cytokine concentrations using Indirect ELISA. Serum/plasma samples were stored in a –20˚C freezer when they were to be examined within 3–10 months; and/or in a –80˚C deep freezer when they were to be analyzed after a longer period (i.e., >10 months). The missing samples were; two patients with death outcomes immediately after admission. Of whom, one after day 0 sample collection and the other before sampling. Twenty patients were unable to attend day 120 follow-up sampling, because, VL endemic areas in southern Ethiopia were very far apart from the treatment center.

## Microscopy and parasite counting/grading

Experienced physicians collected bone marrow or splenic tissue specimens. Two good-quality smears were prepared, fixed for 1 minute in absolute methanol, and stained with Giemsa stain

for 10 minutes. Amastigotes were revealed as oval-shaped cells containing a nucleus, kineto-plast, and pale blue cytoplasm. Amastigotes were counted in 1000 microscopic fields examined under oil immersion (1000x magnification). Two microscopists first counted amastigotes, and averages were obtained. Parasite grades were assigned to average counts on a scale of 0 to 6. A parasite grade of zero meant that both microscopists did not find amastigotes in the 1000 microscopic fields (100% concordance). Grades from 1 to 6 were assigned on a logarithmic scale based on the rule given in S2 Table [26].

## Detection of anti-leishmanial antibody by Indirect ELISA and determination of Antibody Index (AI) values

The ELISA method is based on detecting antibodies in serum samples using 96 well microtiter plates coated with leishmania-derived antigen. Kits were provided by Vircell (Granada, Spain). The test kit detects IgG and IgM antibodies against *L. donovani* in human serum. Test procedures were carried out as per the manufacturer's instructions. A brief description of the procedure is provided in S1 Procedure. Positive and negative sera were included in each plate to standardize the readings and to correct variations. OD values were converted to an "index" of numbers such that they could be given as cardinal numbers signifying antibody levels semi-quantitatively. The formula "(Sample OD/Cut-off OD)*10" was used to generate Antibody Index (AI) numbers.

AI numbers were also translated to qualitative data, and dichotomized as positive (AI numbers > 11), negative (AI numbers < 9), or equivocal (AI numbers 9–11).

## Determination of IL-10, IFN-γ, TGF-β1, and IL-2 concentrations in sera/plasma

Cytokine concentrations in VL patients' serum samples were measured using ELISA (Affymetrix, eBioscience Inc., San Diego, CA, USA) for IL-10, IFN-γ, and IL-2 (Affymetrix, eBioscience Inc., Vienna, Austria) for TGF-β1. A brief description of the procedure is provided in the S2 Procedure. The assays were optimized and run to create standard curves for each cytokine to estimate cytokine concentrations in serum samples. Sensitivity ranges of the assays were 2–300 pg/ml for IL-10; 4–500 pg/ml for IFN-γ; 156.3–10,000 pg/ml for TGF-β1 and 2–250 pg/ml for IL-2. The standard curve was linear over 8 log concentration ranges for each cytokine with correlation coefficient ($r^2$) of 0.998, 0.995, 0.999, and 0.999 for IL-10, IFN-γ, TGF-β1, and IL-2 respectively. Serum cytokine levels were calculated by interpolating the standard curve for absorbance readings (Y-axis = OD value) of test samples calculated from standards of known concentrations (X-axis = Concentration) and results were expressed as picograms of cytokine per milliliters (pg/ml) (S1 Fig).

## Data analysis

Data were entered, checked for completeness, and analyzed by using Microsoft Excel (Office 2013) data analysis and chart tools and Statistical Package for Social Services (SPSS) version 21.0. The socio-demographic characteristics, laboratory parameters, antibody index (AI), and cytokine levels were compared between groups using the nonparametric statistics and, when applicable, the mean, median, interquartile (IQ) range, and 95% confidence intervals were calculated.

Group-wise comparison of average values of cytokines and AI values were performed using nonparametric statistics (Mann-Whitney U test for 2 samples or the Wilcoxon Matched-Pairs Signed-Rank Test ([Kruskal-Wallis 1-way ANOVA for k samples]). Spearman's ($r_s$) rank

correlation was computed to assess associations among cytokine levels, AI, and the corresponding patient demographics and/or laboratory and clinical characteristics and between each cytokine pair. The significance level was set at $p \leq 0.001$ for IL-10 and at $p \leq 0.05$ for antibody index and other cytokines (IFN-γ, TGF-β1, and IL-2).

## Results

### Socio-demographic characteristics of the study population

A total of 48 VL cases who had been attending LRTC during the study period were included. All patients were clinically diagnosed with VL and further confirmed by rK39 (InBios International, Inc. Seattle, WA, USA) serology and/or splenic or bone marrow aspiration. The median age of the participants was 18.5 (IQ range: 10.00–25.00) within the age range of 5–58 years. Higher proportions (n = 38, 79.2%) of the subjects were males with occupation of farming (n = 33, 68%) and residing in the VL endemic area (n = 39, 81.3%), (S3 and S4 Tables). All participants presented with fever at the time of admission and tested negative for HIV. Thirty-five patients (76%) were treated with SSG/PM combination for 17 days, and 11 patients (24%) were treated with SSG alone for 30 days. Upon completion of treatment, the initial cure was assessed by bone marrow or splenic aspirates (S3 and S4 Tables).

### Clinical and laboratory characteristics

The median duration of illness of VL patients was found to be 3.0 months (IQ range: 2.00 −4.25 months). During diagnosis, clinical and laboratory findings showed hematologic disorders, especially a decrease in platelet (median 98,000/mm$^3$; IQ range 65,250– 153,000/mm$^3$); WBC (median 1,900/mm$^3$; IQ range 1,400–3,000/mm$^3$); RBC (median 3.12x 10$^6$/μL; IQ range 2.86–3.51x 10$^6$/μL) and low Hb (median 7.25 g/dl; IQ range 6.35–8.63 g/dl). After completion of treatment, significant improvements were noted in these hematological parameters. When comparing pre-and post-treatment data, the spleen size decreased significantly from a median size of 11.0 cm (IQ range 7.25–15.75 cm) to 5.0 cm (IQ range 1.5–8.0 cm) (S2 Fig and S3 Table).

The distribution and pairwise comparison of pre-and post-treatment time-points indicate statistically significant differences (SSDs) in the absolute counts of WBC, platelet, and measurements of Hb. Compared to baseline values, significantly increased levels of Hb and counts of WBC and platelets were found on day 14, increasing from 1,850/mm$^3$ to 3,250/mm$^3$ for WBC; from 7.2 mg/dl to 8.9 mg/dl for Hb; and from 98,000/mm$^3$ to 255,000/mm$^3$ for platelet counts. Median values at end of treatment (day 18 or 30) were 4,500/mm$^3$ for WBC; 11.3 mg/dl for Hb and 287,000/mm$^3$ for platelet counts. The incremental improvements in the above laboratory parameters showed favorable prognoses; with patients' hematological measurements approaching the normal range around 14 days after treatment initiation (Fig 1 and S3 Table). The mean BMI increased from 16.1 kg/m$^2$ at day 0 to 16.6 kg/m$^2$ at day 18 EOT (PM + SSG) and to 17.1 kg/m$^2$ at day 30 (SSG alone) EOT (Fig 1 and S5 Table).

### Serum levels of anti-leishmanial antibodies

Total anti-leishmanial antibody levels (IgG/IgM) were evaluated before treatment, at different time points during treatment and, 120 days post-treatment. The median AI values at days 0, 7, 14, 18 [EOT], 30 [EOT], and 120 respectively were: 16.0 (IQ range 13.6–19.3); 15.9 (IQ range 13.5–18.1); 16.0 (IQ range 14.1–19.9); 18.3 9 (IQ range 17.6–23.9); 15.5 (IQ range 11.5–17.4); and, 18.7 (IQ range 16.9–19.6 By pairwise comparison using one-way ANOVA (Fig 2 and S6 Table), median AI values at baseline (day 0), during treatment (days 7 and 14) and at EOT

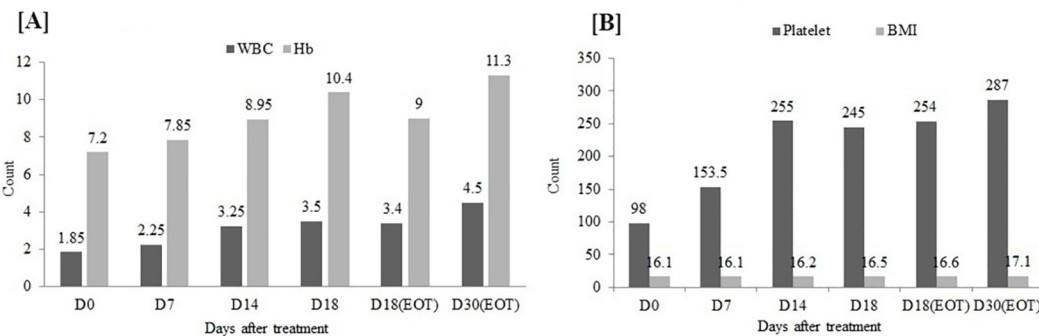

**Fig 1. Laboratory parameters measured before, during, and after treatment.**

(days 18 or 30) were significantly higher (P<0.05) than the median AI values in sera from 20 healthy controls (4.6 [IQ range 3.4–5.7]) or patient samples obtained 120 days after treatment completion (9.6; [IQ range 6.8–12.3]).

## Serum cytokine concentrations of Interleukin 10 (IL-10), Interferon-gamma (IFN-γ), Transforming Growth Factor beta 1 (TGF-β1), and Interleukin 2 (IL-2)

**IL-10.** Higher concentrations of IL-10 were observed in sera of untreated VL patients (n = 48) compared to concentrations in sera of the same patients upon treatment completion and also higher than levels of 20 healthy subjects (Fig 2 and S6 Table). Our data also show a sharp decline in concentrations of serum IL-10 to below the detection limit within seven days of treatment.

The concentrations of IL-10 in sera of VL patients and healthy control subjects (n = 20) were compared using either the Mann-Whitney U test for 2 samples or the Kruskal-Wallis 1-way ANOVA for k samples; and marking SSDs at 0.001. High levels of IL-10 were measured before treatment, with a median concentration of 46.1 pg/ml; (IQ range 16.5–71.7 pg/ml) which sharply decreased with treatment to below detection limit throughout the follow-up schedules (days 7, 14, 18 [EOT], 30 [EOT] and 120) (Figs 2 and 3 and S6 Table); with the exception in one patient whose concentrations increased from 60.7 pg/ml before treatment to 98.30 pg/ml at day 120.

The exceptional observation about serum concentrations of IL-10 in this single patient is striking. The patient was initially treated for 17 days with a combination of SSG and PM for 17

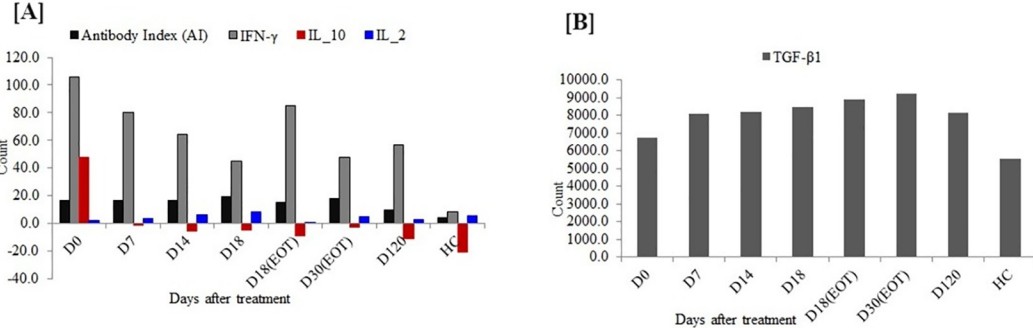

**Fig 2. Antibody Index, IFN-γ, IL-10, IL-2, and TGF-β1 serum concentrations in VL patients before, during, and after treatment cf. concentrations in healthy controls.**

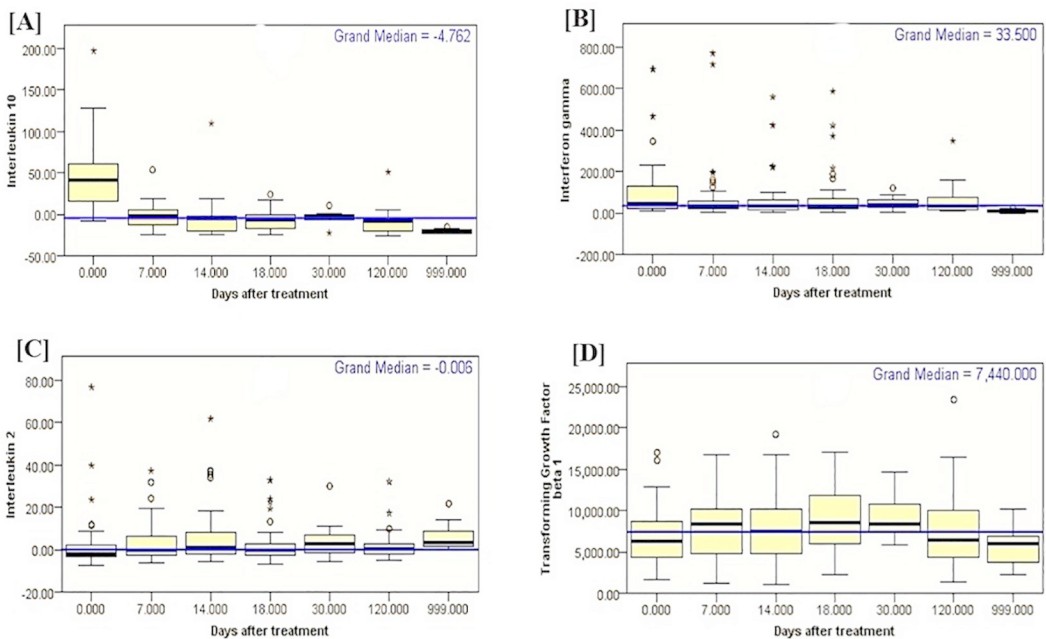

**Fig 3. Box plots showing serum cytokine concentrations of IFN-γ, IL-10, TGF-β1, and IL-2 in pg/ml (Independent sample median tests).**

days, the IL-10 concentration was 60.70 pg/ml before treatment at day 0. The patient had to be re-treated with SSG (30 days, at a dose of 20 mg/kg) as ToC on day 18 revealed a parasite grade of 2+. After 30 days of treatment, the splenic aspirate revealed a sustained parasite grade of 2+, at which point the IL-10 concentration was 98.3 pg/ml. Finally, the patient was treated with amphotericin B (AmBisome) and discharged with microscopy negative results, and that stage IL-10 concentration was 10.48 pg/ml.

**INF-γ.** Serum concentrations of IFN-γ at day 0 (median 47.4 pg/ml; IQ range 21.9− 134.6 pg/ml) was measured in sera of 48 VL patients and in the consecutive post-treatment day 7 (median 33.3 pg/ml; IQ range 19.7−68.5 pg/ml); day 14 (median 35.2 pg/ml; IQ range 16.0 −65.9 pg/ml); day 18 (median 33.4 pg/ml; IQ range 26.2−56.5 pg/ml); day 18 EOT (median 43.0 pg/ml; IQ range 19.3−80.3 pg/ml); day 30 EOT (39.5 pg/ml; IQ range 28.8−78.3 pg/ml); and, day 120 (median 33.5 pg/ml; IQ range 13.3−74.1 pg/ml). IFN-γ serum concentration of healthy controls (n = 20) had median and IQ ranges of 8.0 pg/ml; (2.6−10.7 pg/ml) (Figs 2−4 and S6 Table). Concentrations of IFN-γ before treatment and in consecutive post-treatment serum samples of VL patients were significantly ($p < 0.001$) higher than plasma samples of healthy controls at 0.05 (Kruskal-Wallis 1-way ANOVA for k samples).

**TGF-β1.** Median concentrations of TGF-β1 in sera of VL patients were 6215.0 pg/ml (IQ range 4090.0−8690.0 pg/ml) at day 0; 8390.0 pg/ml (IQ range 4815.0− 10315.0 pg/ml) at day 7; 7540.0 pg/ml (IQ range 4740.0−10465.0 pg/ml) at day 14; 8515.0 pg/ml (IQ range 7165.0− 10365.0 pg/ml) at day 18; 8340.0 pg/ml (IQ range 5927.5−13040.0 pg/ml) at day 18 EOT; 8365.0 pg/ml (IQ range 7265.0−11665.0 pg/ml) at day 30 EOT; and 6365.0 pg/ml (IQ range 4290.0−10540.0 pg/ml) at day 120. The corresponding median concentrations in serum of healthy controls were 5990.0 (IQ range 3752.5−6840.0 pg/ml) (Figs 2 and 3 and S6 Table). The pairwise comparison of TGF-β1 serum concentrations using Kruskal-Wallis 1-way ANOVA didn't indicate SSDs at ($p \leq 0.05$) across the various post-treatment schedules in VL patients.

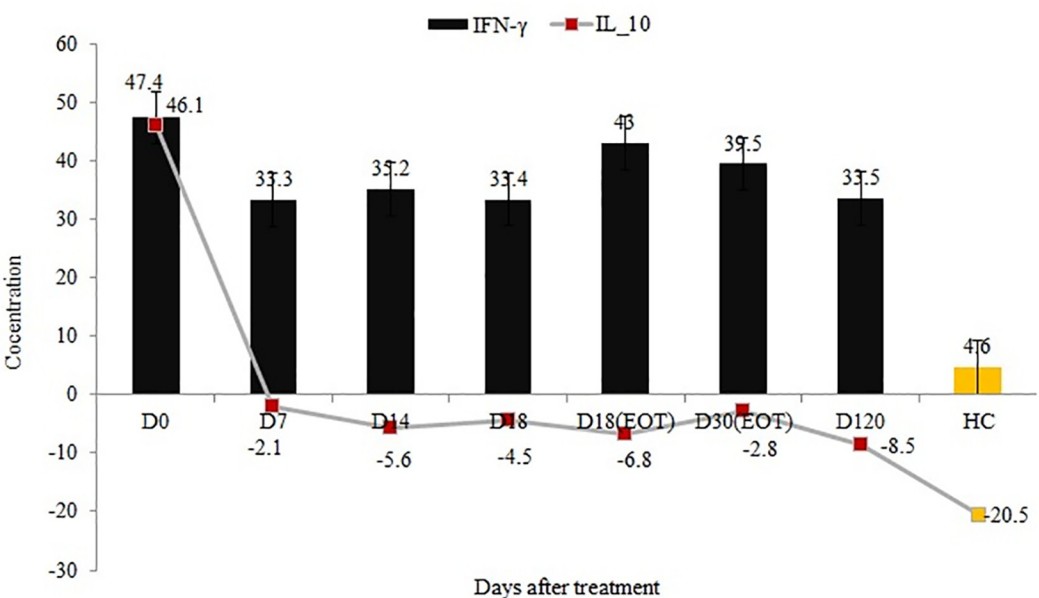

**Fig 4. IFN-γ gamma and IL-10 serum cytokine concentrations in VL patients before and after treatment cf. concentrations in healthy controls.**

**IL-2.** Lower (below detection limits) concentrations of IL-2 at day 0 (median concentration of -1.8 pg/ml; IQ range -3.4–3.2 pg/ml) were measured in serum samples of 48 VL patients and in consecutive post-treatment of day 7 (median -0.1 pg/ml; IQ range -2.8–7.1 pg/ml); day 14 (median 1.0 pg/ml; IQ range -2.1–8.4 pg/ml); day 18 (median 0.3 pg/ml; IQ range -2.0 −24.0 pg/ml); day 18 EOT (median -1.1 pg/ml; IQ range -3.4–1.9 pg/ml); day 30 EOT (median 2.6 pg/ml; IQ range -1.9–8.1 pg/ml); and, day 120 (median 0.6pg/ml; IQ range -2.2–3.9 pg/ml). Concentrations of IL-2 in 20 plasma samples of healthy controls (median 3.5 pg/ml and IQ ranges 1.2–9.3 pg/ml), which was a bit higher than concentrations in sera of VL patients (Figs 2 and 3 and S6 Table).

## Comparison of parasite grades and serum cytokine concentrations

Levels of IL-10, IFN-γ, TGF-β1, and IL-2 were measured in serum samples of VL patients (n = 48) at the baseline before treatment and at EOT. Patients with higher concentrations of serum IL-10 had higher counts/grades of parasites before the initiation of treatment (Fig 5). Serum concentrations of both IL-10 and IFN-γ were significantly higher in patients with active VL (Fig 4). IL-10 concentration was found to correlate strongly (r = 0.56) and significantly ($p \leq 0.001$) with splenic aspiration parasite grade. The overall correlation, however, was found to be weak for the other three cytokines tested (r = 0.07 for IFN-γ, 0.21 for TGF-β1, and 0.09 for IL-2). The median serum concentration of IL-10 at baseline in patients with different parasite grades (PG) were 52.3 pg/ml (at 1+ PG), 31.4 pg/ml (at 2+ PG), 24.1 pg/ml (at 3+ PG), 48.5 pg/ml (at 4+ PG), 59.0 pg/ml (at 5+ PG), and 37.2 pg/ml (at 6+ PG) (Fig 5A and 5B).

The median serum concentrations of IFN-γ in patients diagnosed with different parasite grades were found to be 67.5 pg/ml (at 1+ PG), 18.8 pg/ml (at 2+ PG), 161.1 pg/ml (at 3+ PG), 35.1 pg/ml (at 4+ PG), 38.8 pg/ml (at 5+ PG), and 45.4 pg/ml (at 6+ PG) (Fig 5A and 5B). There was a tendency of higher median serum concentrations of IFN-γ being measured at lower parasitemia, e.g., at 1+ and 3+ (Fig 5C).

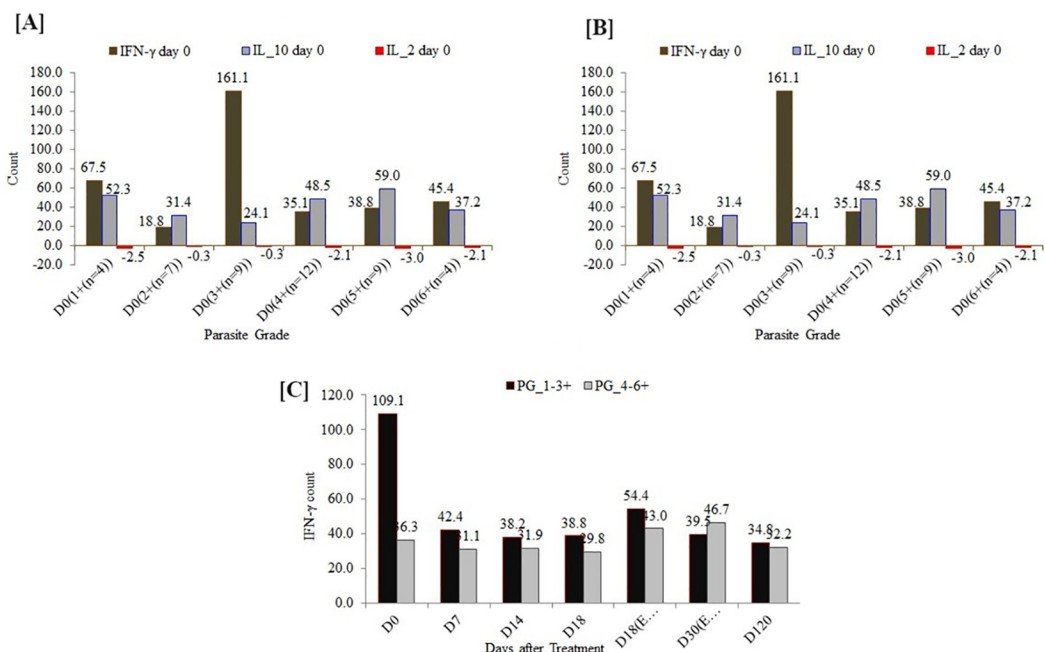

**Fig 5. Comparison of the baseline parasite grade (PG) and serum cytokine concentrations.**

### Comparison of parasite grade and levels of anti-leishmanial antibody

A high level of anti-leishmania antibodies (IgG/IgM) was measured in all VL patients regardless of parasite grade before treatment and at a different schedule during the treatment (Fig 2 and S6 Table). The median of AI values at day 0 did not show SSDs ($p \geq 0.05$) between any categories of parasite grade (Independent samples median test and/or Kruskal-Wallis Tests); thus, no multiple comparisons were performed. However, the difference in AI values of healthy controls showed SSDs ($p < 0.05$) even when comparison was made in serum samples of patients obtained after 120 days (Fig 6). Comparison of hematologic parameters such as WBC and HB; and clinical like BMI and spleen size with levels of anti-leishmanial antibody IgG/IgM (AI) and cytokine concentrations of IFN-gamma, IL-10 and IL-2 are shown in S2 and S3 Figs.

## Discussion

The present study investigated the correlation among different cytokine concentrations, and antibody levels in serum samples, clinical and laboratory parameters associated with VL patients and assessed their interplay to explore the time-dependent alterations of their immune components in defining the disease presentation and patient outcomes in response to treatment. Interestingly, these results revealed a strong correlation of increased IL-10 with different clinical and laboratory parameters, implying disease manifestations to be linked with the increased immunological markers as well as clinical-like body-mass-index, splenomegaly, and pancytopenia (S2 and S3 Figs). Herewith, we studied 48 patients with the presentation of active VL by evaluating the splenic/bone marrow aspirate parasitic grade, cytokine concentrations, and antibody levels before treatment, during treatment at different times, and post-treatment to identify biomarkers of disease state and to assess responses to treatment. It was implicated that cytokines may be used as biomarkers to monitor VL and treatment follow-up's clinical progression [27].

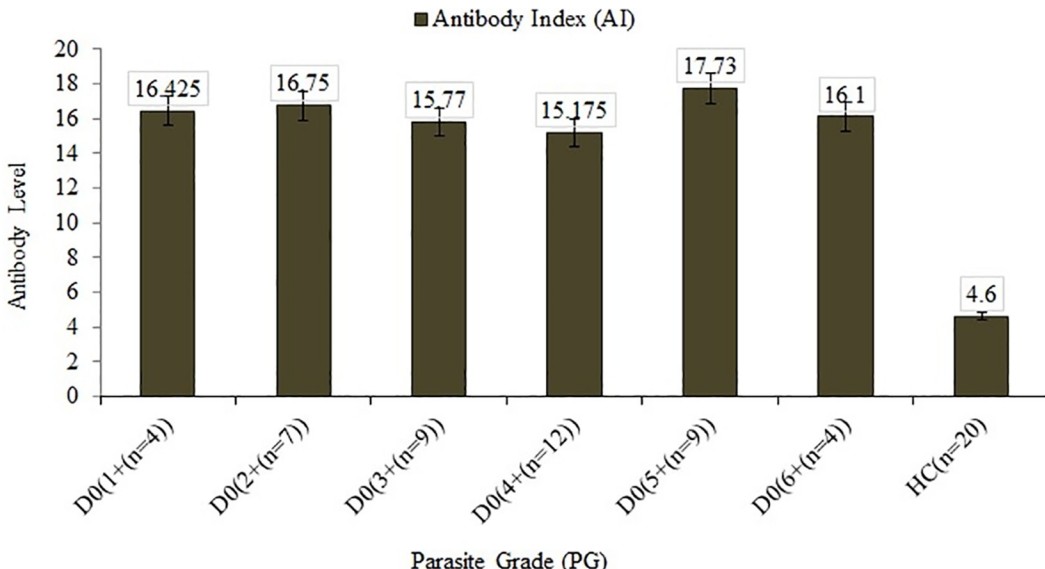

**Fig 6. Comparison of parasite grade and anti-leishmanial antibody levels measured with Indirect ELISA.**

In this study, the serum levels of IL-10, IFN-γ, and TGF-β1 were higher in sera of active VL patients than healthy controls (Figs 2 and 3). Serum concentrations of IFN-γ and IL-10 were reported to be elevated in VL [20]. It was also reported that plasma concentrations of IL-10, IL-12, and IFN-γ, were higher in patients than in family members and control individuals [28] implying that disease processes following infection are important to induce immune activation and cytokine production. Interestingly, base-line serum concentration of IFN-γ was higher among patients diagnosed with lower parasite load cf. those with higher parasite load (Fig 5C). This observation illustrates the role of Th1 responses in the control of VL parasitemia. Consequence of this tempting to hypothesize that indeed high concentration of IFN-gamma during diagnosis of VL may be a marker of a good prognosis. On the contrary, our result showed the level of IL-10 was high during active VL regardless of parasite grade. IL-10 has been well known for its role in VL pathogenesis and a significant correlation was already reported between parasite load and serum IL-10 levels [29]. The abundance of *Treg* cells, along with IL-10 and TGFβ1, in VL patients, may be the main reason for large numbers of parasites inflicting disease aggravation [30]. TGF-β1 has been shown to impart down-regulatory effects on macrophages and its blockage could restrict parasite replication [31]. Previous studies had documented elevated levels of IL-10 in VL patients [11,20,27,29,32] and its association with pathology. Thus, IL-10 is well established as a critical component in Leishmania donovani infection, contributing to the sustained pathology seen in untreated patients. During treatment, IL-10 concentrations declined sharply, while IFN-γ levels were sustained for longer periods before they started to decrease substantially (Fig 4). This observation indicates, after treatment, Th1 activation is sustained for a significant period after IL-10 levels have fallen sharply. The interplay between parasitemia, IL-10, IFN-gamma, and other mediators is important in determining patient outcomes during treatment, particularly in the control of pathological inflammatory responses and in the ultimate acquisition of immunological memory and protective immunity. Clinical immunological studies of this type if conducted in larger scope and sample size may indeed shed light on the kinetics and regulation of IL-10 and IFN-γ. In this study, the observed sharp fall in IL-10 levels following the start of treatment within seven days is coincident with the control of parasite multiplication, lending support to an important role of this cytokine in human VL. As pointed out above, while IL-10 concentrations decreased sharply to below detection limit

at day 7 and throughout the consecutive follow-up schedules, the decline in IFN-γ concentrations and antibody levels (IgG/IgM) lagged until 120 days post-treatment at which stage the reductions were statistically significant. In a previous study by Caldas et al. [32], the high levels of IL-10 and IFN-γ observed during active VL decreased in most patients by 30 days post-treatment and almost vanished 120 days after treatment.

Production of IL-10 has been suggested to be important for the survival and persistence of parasites inside macrophages [25,27]. However, IL-10 protects against the side effects of exaggerated inflammatory responses [17,18].

As a self-regulating feedback mechanism, many pro-inflammatory molecules can induce regulatory responses. Parasite-induced IFN-γ, in combination with the immune complex formation and other macrophages, promoted mediators, may facilitate the generation of IL-10 producing T cells and block generation of potentially protective Th17 cells [24].

The clinical outcome of the disease is a consequence of the complex interaction between the pathogen and the host, and the survival of the pathogen largely depends on the type and balance of immune effectors and regulatory responses being produced by host immune cells [29]. We, therefore, measured serum concentrations of cytokines previously associated with Th1/Th2 or anti-/inflammatory responses as well as antibodies aiming at identifying prognostic markers. We were fortunate to have encountered a patient who had come back with VL relapse 120 days after treatment. The patient presented with high serum concentrations of IL-10, IFN-γ, TGF-β1, as well as high titers of anti-leishmanial antibody. Interestingly, these elevations during initial diagnosis/treatment and after treatment were also increased. Albeit in one patient, this data gives anecdotal evidence that these cytokines (IL-10, IFN-γ, and TGF-β1) might be useful biomarkers of prognosis and could be used to assess responses to treatment. Due to the dynamic kinetics of IL-10 during treatment of VL, prognostication by IL-10 might prove to have a clinical significance.

For effective control of VL in endemic areas; rapid and reliable diagnostic and prognostic tools are desperately needed. A combination of non-invasive case detection by rapid diagnostic tests such as rk39 based lateral flow assays, combined with a test of cure biomarker such as IL-10 can improve monitoring of VL therapy and disease outcome.

## Conclusion

This study has combined cytokine assays with parasitemia assessments and antibody assays in a well-designed descriptive study. Regardless of microscopic parasite load and levels of antibody; elevated levels of IL-10 in VL patients' serum significantly decreased to below the detectable limit after seven days of treatment. These results suggest that IL-10 could be used as a marker of prognosis and possibly in the prediction of long-term patient outcomes. Further, we noted that high concentrations of INF-γ were associated with lower parasitemia at baseline assessment and that levels of IFN-γ were sustained during the follow-up period. These data suggest that the interplay of Th1 and Th2 responses in VL patients are valid in the understanding immunopathogenesis of visceral leishmaniasis. Even though this is a well-designed study that examined the relationships between clinical presentation, parasitemia, immune progression, response to treatment, and parasite clearance, it was limited in the array of mediators selected. Investigations of other mediators, e.g. IL-4, IL-5, IL-6, IL-12, IL-18, TNF-α, and others have not been included. Future studies will help to identify additional immune mediators of pathogenesis and biomarkers of prognoses.

## Supporting information

**S1 Fig. Standard curves of each cytokine concentrations.**
(DOCX)

**S2 Fig. Spleen size of VL patients at day 0 and end of treatment with various grades of parasitemia.**
(DOCX)

**S3 Fig. Comparison of hematologic and clinical parameters with levels of anti-leishmanial antibody and cytokine concentrations.**
(DOCX)

**S1 Table. Residence and travel history of study participants.**
(DOCX)

**S2 Table. Parasite grading procedures in Giemsa-stained smears of splenic /bone marrow tissue.**
(DOCX)

**S3 Table. Description of the study population by socio-demographic characteristics, and clinical/laboratory parameters at baseline and EOT.**
(DOCX)

**S4 Table. Description of the study population by socio-demographic characteristics, and clinical/laboratory parameters at baseline and EOT (Median and IQ Range) shown in children (age 5–17) and adults (>17 years).**
(DOCX)

**S5 Table. Comparison of the study population's clinical/laboratory parameters shown by treatment regimen administered.**
(DOCX)

**S6 Table. Mean/median AI values of IgG/IgM and cytokine concentrations.**
(DOCX)

**S1 Procedure. ELISA Procedure for measuring of anti-leishmanial antibody.**
(DOCX)

**S2 Procedure. ELISA procedure for measuring cytokine concentrations (IL-10, IFN-γ, TGF-β1, and IL-2).**
(DOCX)

**S1 Data. Supplementary excel file.**
(XLSX)

## Acknowledgments

The authors acknowledge all the VL patients for their cooperation to have consented and for their participation in the study. We are grateful to staff members of Arba Minch Hospital LRTC for their assistance during sample and data collection. Special thanks to Dr. Tamiru Shibiru, Dr. Kusia Ayano, Tedla Teferi, Solomon Getu, Abera Adisu, and Ephrem Ayele.

## Author Contributions

**Conceptualization:** Dagimawie Tadesse, Asrat Hailu.

**Data curation:** Dagimawie Tadesse, Alemseged Abdissa, Asrat Hailu.

**Formal analysis:** Dagimawie Tadesse, Alemseged Abdissa, Mekidim Mekonnen, Tariku Belay, Asrat Hailu.

**Investigation:** Dagimawie Tadesse, Asrat Hailu.

**Methodology:** Dagimawie Tadesse, Alemseged Abdissa, Mekidim Mekonnen, Tariku Belay, Asrat Hailu.

**Project administration:** Dagimawie Tadesse.

**Resources:** Dagimawie Tadesse, Asrat Hailu.

**Software:** Dagimawie Tadesse, Alemseged Abdissa, Mekidim Mekonnen, Asrat Hailu.

**Supervision:** Dagimawie Tadesse, Alemseged Abdissa, Mekidim Mekonnen, Tariku Belay, Asrat Hailu.

**Validation:** Dagimawie Tadesse, Alemseged Abdissa, Tariku Belay, Asrat Hailu.

**Visualization:** Dagimawie Tadesse, Alemseged Abdissa, Asrat Hailu.

**Writing – original draft:** Dagimawie Tadesse, Asrat Hailu.

**Writing – review & editing:** Dagimawie Tadesse, Alemseged Abdissa, Mekidim Mekonnen, Tariku Belay, Asrat Hailu.

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
