## [Decision Letter · Decision Letter 0]

12 Jan 2021

Dear Mr Abate,

Thank you very much for submitting your manuscript "Antibody and cytokine levels in visceral leishmaniasis patients with varied parasitemia before, during and after treatment in patients admitted to Arba Minch General Hospital, southern Ethiopia" for consideration at PLOS Neglected Tropical Diseases. As with all papers reviewed by the journal, your manuscript was reviewed by members of the editorial board and by several independent reviewers. In light of the reviews (below this email), we would like to invite the resubmission of a significantly-revised version that takes into account the reviewers' comments. 

We cannot make any decision about publication until we have seen the revised manuscript and your response to the reviewers' comments. Your revised manuscript is also likely to be sent to reviewers for further evaluation.

Sincerely,

Susan M Bueno

Associate Editor

Charles Jaffe

Deputy Editor

Reviewer's Responses to Questions

**Key Review Criteria Required for Acceptance?**

**Methods**

-Are the objectives of the study clearly articulated with a clear testable hypothesis stated?

-Is the study design appropriate to address the stated objectives?

-Is the population clearly described and appropriate for the hypothesis being tested?

-Is the sample size sufficient to ensure adequate power to address the hypothesis being tested?

-Were correct statistical analysis used to support conclusions?

-Are there concerns about ethical or regulatory requirements being met?

Reviewer #1: Analyzes of cytokines such as IL-12 and TNF-alpha could be included in this study.

Reviewer #2: Please see summary and general comments section

**Results**

-Does the analysis presented match the analysis plan?

-Are the results clearly and completely presented?

-Are the figures (Tables, Images) of sufficient quality for clarity?

Reviewer #1: (No Response)

Reviewer #2: Please see summary and general comments section

**Conclusions**

-Are the conclusions supported by the data presented?

-Are the limitations of analysis clearly described?

-Do the authors discuss how these data can be helpful to advance our understanding of the topic under study?

-Is public health relevance addressed?

Reviewer #1: Future analyzes should not be included in the conclusion.

Reviewer #2: Please see summary and general comments section

**Editorial and Data Presentation Modifications?**

Reviewer #1: Minor Revision: 

lines 34-35: AI levels decrease in healthy controls?

line 37: clinical-epidemiological

Introduction: better describe the role of cytokines related to this study (IL-2, TGF-beta and IFN-gamma).

line 200: Fig 1 could be shown in the supporting information.

line 241: change to 76%

Table 2: review the "variables" column (EOT)

Reviewer #2: (No Response)

**Summary and General Comments**

Reviewer #1: (No Response)

Reviewer #2: This interesting study titled : ”Antibody and cytokine levels in visceral leishmaniasis patients with varied parasitemia before, during and after treatment in patients admitted to Arba Minch General Hospital, southern Ethiopia.” from Ethiopia, is easy to learn, and well presented. 

For the first and quick read it seams that the main focus is to provide the correlations between Antibody and cytokine with depend of the level of VL parasitemia in different steps linked to receiving treatment. The authros find that IL-10, IFN-γ, and TGFβ could be the markers that help to distinguish the active VL form to the latent or the asymptomatic ones and the concentration of cytokines could be a criteria of cure level. 

Hereby some suggestions that could enhance the advanced presentation of this manuscript raised after full reading process.

1- Our approach is to understand how the immune response generated during different treatment stages of active VL before, during and post-treatment that are impacted by the presence of different cytokines resulting in clinical pathogenesis and disease.

This sentence could be better if replaced by the one below 

Our approach is to understand how the immune response generated during different treatment stages of active VL before, during, and post-treatment is impacted by the presence of different cytokines resulting in clinical pathogenesis and disease.

2- …. found in areas of L. donovani transmission >>>> found in L.donovani transmission areas …

3- In Ethiopia, with an estimated 4,500 to 5,000 new cases occur every year, the population at risk is more than 3.2 million

>>>Could be reformulated with an easier way 

4- in the Eastern Africa Region, new challenges of VL control attributed to lack of appropriate tools and HIV co-infection have eventually led to resurgence of leishmaniasis and its spread to new geographical areas.

>>> what is the meaning of “lake of appropriate tools” ? Please clarify this sentence or remove it. (The introduction could be shortened by keeping only the essential information in consistence with the research question)

5- Untreated VL patients act as reservoirs for parasites and therefore contribute to disease transmission in anthroponotic VL areas

>>>> please provide a good reference that explain clearly the anthroponotic VL transmission (Directly from human to human by only a phlebotomy or another vector) in Ethiopia or elsewhere. 

6- In the introduction section, the authors explain the usefulness of a test of VL cure in the context of the HIV coinfected patients. However, the 48 patients selected for the study did not have any detailed information about their HIV condition. Please provide the results based on this criterion. Or focus of the manuscript on the immunocompetent patients. The only information is about POSITIVITY OF THE TEST but not the immunodeficiency state which interact with the antibody levels 

7- Just the next paragraph, the authors explain the importance of asymptomatic VL persons to be detected>>> same remark for the study. Only Active VL was selected. 

8- The paragraphs that include the presentation of markers and their link to VL and the immunity need a better reformulation by focusing on the main questions of this study. (What could be remembered is that the authors are formulating questions about the role of VL treatment and they presented only the increase of markers in case of active VL what about the asymptomatic and the HIV infected persons, what about the provided treatment …etc)

9- “Concerning prognostic tools, it is a worthful effort testing if concentrations of selected cytokines in serum could be used to monitor treatment outcomes.”

>> it could be better to start with this reflexion and find relevant references to explain which kind of cytokines. 

10- The study is a clinical laboratory study that need to be presented based on an appropriate guideline and checklist (Please find the suitable one in the EQUATOR network website).

11- The results could be compared to the accuracy of the rk39 rapid test used. Is it one rapid diagnostic test (As cited InBios*) or different kind of trademark? In case of many trademark what are the known accuracy and how the authors re-confirmed the diagnostic. 

12- This is really “A prospective cohort study” Definitely not. What is described is the way to get the blood samples to start the laboratory study. (May be the authors intend to present another manuscript putting on the spot such prospective cohort study) But for the actual manuscript focusing on the laboratory results is the best strategy.

13- If already the study involving the recruited patients as cited “ A prospective cohort study was conducted among admitted VL patients who were treated and followed at different time points between 02/September/2015 to 01/August/2016” >>> Please provide the reference and focus on the origin of the blood samples. 

14- All this information should be shown as a supplementary material and presented as table and combine in excel form all the information provided in S1 table. There is a need to show who take the combination or the monotherapy (combination treatment of Paromomycin Sulphate (PM) and Sodium Stibogluconate (SSG) or day 30 for those who treated with Sodium Stibogluconate (SSG) monotherapy) The results normally will be presented based on this difference of received treatment. 

15- There is a need of precising HIV conditions of the patients? If the information is available to be consistent with the reflexion in the introduction section. 

16- For this study, it’s important to precise how the samples were managed stored in which conditions and if there is any messing data with regard the no possibility to get all blood sampling in the targeted days. 

17- Since 1990s, ample evidence has been generated that a combination of the WHO clinical case definition for VL and a positive antibody test is an adequate for decision to treat [22,23]. >>>> There is no need for such explanation. Or you can move it to an appropriate place in the introduction section. 

18- How many technicians or may be the authors involved to count and put the index for microscopy. The graded scale was used with only one and same person; what is the degree of coherence between two persons who did this grading. Please add kappa coefficient and explain if any pictures of the amastigots L.donovani forms were taken and how the slides were stored. Could you add pictures for each of the 6 grades as Table (Figure)S2

19- To shorten the Method section please add the full technic to analyse Indirect ELISA for detection of anti-leishmania antibody and the Cytokine Profile (IL-10, IFN-γ, TGF-β1, and IL-2) as a supplementary material and keep just some words in the method section. 

20- For the ethical clearance, if the study is already published elsewhere, is important to mention if the authors where allowed to take the stored blood samples for doing this study

21- As the legal age is 18 y o Please provide the age range between 5 and 17yo in replacement of the sentence (the age ranges of 5-58. Higher proportions (79.2%)) 

22- If the children represent almost the half of the blood samples, please explain in the discussion part if the biomarkers targeted had different concentrations based on the age group of the normal population without including the VL condition. Just to be sure that the potential confounding factors are neutralized. 

23- What is the reason of treating some patient with a combined ttt and the others less than 24% by a monotherapy? All patients were hospitalized by the duration of the hospitalisation is not noticed (please in the Future Excel file it could be very informative to add the ages and the treatment provided, doses and duration of the hospitalisation, with the concentration levels of the laboratories parameters for each of the 48 blood samples analysed. 

24- TABLE 1 does not give any important information by seeing Median and interquartile statistics to help understanding the correlation between the socio demographic characteristics and the level of parasitemia or the level of Interferon gamma or the cytokines. A better information could be provided with a p value to show which socio demographic characteristics could or not influence the laboratory parameters or Microscopy grade parasetemia in each targeted date (D0, D7 D18 …D120) (The small sample will be managed with fisher exact analysis)

25- Same remark for all the rest of the presented information in the result section. Too many that need to be synthetised with remind that in general for one manuscript 5 tables or Figures should be on the main text to be attractive and easy to read. 

The main problem in the presentation of the results is the absence of the standardised comparison. Based on the introduction section, the main objective is to show active vs asymptomatic cases (is not possible from the collected data) the other main objective is to present HIV positive and HIV negative cases and the influence of such condition on the results. Another main objective is the before and after treatment results. The other main information is based on the potential Cytokines concentration to decrease with the cure state. 

One of the propositions to readdress the heavy information in the result section is to focus primarily on the cure test, which is based on the initial bone marrow parasitological grade in Microscopy and the final bone marrow parasitological grade in the microscopy (after getting all the treatment) The treatment effectiveness is the first information that could be analysed. 

Then the comparison on the laboratory markers change before starting the treatment during the treatment and after the treatment become interesting to provide in very concise way. 

Secondly, on the concentration change of the cytokines compared to the different groups of comparison ttt, hiv condition, sympt vs asympto, and the main socio demographic characteristics (which is not possible as all the 48 cases are only active VL cases, and the HIV condition is not available) 

26- In the discussion section, is not relevant to cite all these paragraphes :

For the clinical aspects, the high prevalence of palpable spleen and severe cytopenia among true VL cases with rk39 positive tests are the most significant aspects that may increase clinical suspicion of VL. The recombinant antigen, rK39 is specific for antibodies against VL caused by L. donovani complex members. It is highly sensitive and predictive for the onset of acute disease [26]. Although the sensitivity of the test is high, it is entirely influenced by the antigen used in the test [10]. In contrast, rK39 does not show detectable antibodies in cutaneous or mucocutaneous leishmaniasis [27]. In this data, all patients had palpable spleen range from 3 to 24cm with a median size of 12cm and 100% of the participants showed rk39 rapid test positive result. 

27- As the actual study used just active VL cases, it’s also not relevant to compare with the other Ethiopian study when the asymptomatic cases were investigated “This is in agreement with a study in south-western Ethiopia, serum cytokines (IFN-γ, IL-12, and IL-15) in symptomatic VL patients were significantly higher than in patients with asymptomatic Leishmania infection and healthy controls with significant decrease of IFN-γ and all mediators observed after treatment [12,19]. »

28- Please rethink the added value of this study in comparison of what is already known. Such “base-line serum concentration IFN-γ was higher among patients diagnosed with lower parasite load cf. those with higher parasite load” and the cytokines results. is there any difference between man and woman, age groups less of 18 yo and older, HIV test result, the package including the whole definition of a confirmed case of VL compared to the ones who have less symptoms and laboratory threshold levels? And the duration of illness before being confirmed as VL or being visiting conventional health structures. 

29- There is no section about the limits of the study. Moreover, some sentences in the conclusion part could be moved to such future limits of the study section 

Sure, the next version will be more concise, particularly focused on the main messages and reply directly to the valuable reformulated research questions decided by the authors of this interesting and hard work.

PLOS authors have the option to publish the peer review history of their article (what does this mean?). If published, this will include your full peer review and any attached files.

Reviewer #1: No

Reviewer #2: No
---

## [Decision Letter · Decision Letter 1]

2 May 2021

Dear Mr Tadesse,

Thank you very much for submitting your manuscript "Antibody and cytokine levels in visceral leishmaniasis patients with varied parasitemia before, during and after treatment in patients admitted to Arba Minch General Hospital, southern Ethiopia" for consideration at PLOS Neglected Tropical Diseases. As with all papers reviewed by the journal, your manuscript was reviewed by members of the editorial board and by several independent reviewers. The reviewers appreciated the attention to an important topic. Based on the reviews, we are likely to accept this manuscript for publication, providing that you modify the manuscript according to the review recommendations. 

Sincerely,

Susan M Bueno

Associate Editor

Charles Jaffe

Deputy Editor

Reviewer's Responses to Questions

**Key Review Criteria Required for Acceptance?**

**Methods**

-Are the objectives of the study clearly articulated with a clear testable hypothesis stated?

-Is the study design appropriate to address the stated objectives?

-Is the population clearly described and appropriate for the hypothesis being tested?

-Is the sample size sufficient to ensure adequate power to address the hypothesis being tested?

-Were correct statistical analysis used to support conclusions?

-Are there concerns about ethical or regulatory requirements being met?

Reviewer #1: There are no major revision request.

Reviewer #2: The objectives of the study were clearly articulated, the population was clearly described and correct statistical analysis were used. 

The ethical consideration and confidentiality of the participants were guaranted.

**Results**

-Does the analysis presented match the analysis plan?

-Are the results clearly and completely presented?

-Are the figures (Tables, Images) of sufficient quality for clarity?

Reviewer #1: Results are clearly demonstrate.

Reviewer #2: The six figures (Fig 1 to Fig 6) are well described into the results section

**Conclusions**

-Are the conclusions supported by the data presented?

-Are the limitations of analysis clearly described?

-Do the authors discuss how these data can be helpful to advance our understanding of the topic under study?

-Is public health relevance addressed?

Reviewer #1: Conclusions are relevant.

Reviewer #2: The conclusions are suppported by the data presented and the discussion is helpful for more research insight about this topic

**Editorial and Data Presentation Modifications?**

Reviewer #1: accept

Reviewer #2: Sample of small editorial modifications :

Ln 45 The disease is difficult to diagnose >>>> is better to use the condition rather than the disease 

Ln 46-47 We investigated how the immune response generated during different stages of active VL before, during, and post-treatment was influenced by the presence of different cytokines. >>>>> different is repeated twice in the same sentence. 

Ln 83-85 disease and progression [15]. The production of TGF-β by infected macrophages is associated with inhibition of IFN-γ production, suppression of macrophage activation, and progression of disease processes. >>> progression could be replaced by advancement 

Ln 172 to correct plate to plate & day to day variations. >>>> May suggest a short sentence like : to correct variations. 

Ln 368-369 while IFN-γ levels were sustained for longer periods before it started to decline substantially >>> decrease rather than deciline 

Ln 381 at which stage the declines were statistically significant >>>> reductions rather than declines 

Ln 399-400 these elevations during initial diagnosis/treatment and after treatment were also high. >>>> increased rather than high

Ln 459 S1 Fig: Spleen size of VL patients at day 0 and end of treatment with varied grades of parasitemia at day 0.>>> various rather than varied 

Ln 115-118 The study population comprised of all VL cases who would be diagnosed based on composite diagnostic criteria [CoDC] (i.e., clinical or serological diagnosis combined with favorable therapeutic response or parasitology by smear and/or culture). >>>> .. comprised all VL cases ..........

reformulate for more clarety the sentence "combined with favorable therapeutic response or parasitology by smear and/or culture"

Ln 119 the definitive diagnosis was by either of the following >>>> please reformulate 

Ln 135 Prior to studyenrollment >>>> Before the study enrollement 

Ln 139 -141 Following preliminary positive screening test by rk39, clinical and haematological investigations as well as direct microscopic examination of splenic or bone marrow aspiration were done systematically. >>>> Following a preliminary positive screening test by rk39, clinical and haematological investigations and direct microscopic examination of splenic or bone marrow aspiration were done systematically. 

Ln 145 Finding of Leishman Donovan bodies in in tissue aspirates >>>> reformulate please 

Ln 147 were also captured, which include testing for HIV >>>> ... including testing ...

Ln 148 -150 After obtaining informed consent, 7.5 ml of whole blood was also collected by venepuncture in a plain and sodium-EDTA vacutainer (Becton, Dickinson, UK) from each study subject. >>>> Please reformulate the sentence for more clarety and easy understanding 

Ln 150 from each study subject. >>>> the participant or the control blood subject? to clarify please within the sentence if possible. 

Ln 155 Bone marrow or splenic tissue specimens were collected by experienced physicians. >>>> Experienced physicians collected bone marrow or splenic tissue specimens. 

Ln 159-160 Amastigotes were first counted by two microscopists, and averages obtained. >>>> Two microscopists first counted amastigotes, and averages obtained. 

Ln 167 The ELISA method is based on the detection of antibodies in serum samples >>>> .....on detecting antibodies ...

Ln 183 to enable estimation of cytokine >>> to estimate cytokines 

Ln 204 The level of significance >>> The significance level 

Ln 213 Examination of patients’ and specimen collections were performed as per the national guidelines and procedures of Ethiopia. >>>> .... were performed as recommanded by the national guidelines and procedures. 

Ln 346-347 It was implicated that cytokines may be used as biomarkers to monitor the clinical progression of VL and treatment follow up >>>> . It was implicated that cytokines may be used as biomarkers to monitor VLand treatment follow-up's clinical progression 

Ln 357 It is thus >>>>????

**Summary and General Comments**

Reviewer #1: (No Response)

Reviewer #2: The authors replied point by point to all suggested corrections and previous comments. 

The new version of this manuscript is well enhanced and only few remarks remain questionnable :

1- 

In the section (inserted at the end of the manuscript , "Strengths and Limitations:

Even though this is a well-designed study that examined the relationships between clinical presentation, parasitaemia, immune progression, response to treatment and parasite clearance, it was limited in the array of mediators selected. Investigations of other mediators, e.g. IL-4, IL-5, IL-6, IL-12, IL-18, TNF-α and others have not been included. Future studies will help to identify additional immune mediators of pathogenesis and biomarkers of prognoses."

>>> the authors are invited to take only the main message and insert it before the conclusion section. 

2-

in the study participants and sampling section:

A total of 48 patients admitted in LRTC for VL treatment who fulfilled the inclusion criteria were enrolled consecutively. Patients were treated either for 17 days if they received a combination treatment of Sodium Stibogluconate (SSG) and Paromomycin (PM) or for 30 days if they received SSG monotherapy. Patients were followed till end of treatment (EoT) and samples were collected before treatment at day 0, during treatment at days 7 & 14, at EoT (day 18 or 30), and 120 days after EoT. The Ethiopian treatment guideline recommends a 17-day combination treatment of SSG (20mg/kg b.wt) and PM (15mg/kg b.wt). During this study period 11 patients received a 30-day dose of SSG as paromomycin was not available for all patients. 

>>>> please reformulate the treatment administred like for example:

A total of 48 patients admitted in LRTC for VL treatment who fulfilled the inclusion criteria were enrolled consecutively. the mean treatment provided was the one recommanded by the Ethiopian MoH guideline combining (20mg/kg b.wt) of Sodium Stibogluconate (SSG) and (15mg/kg b.wt) of Paromomycin (PM) for 17 days. However, due to shortcut of PM during the study period 11 patients from the 48 ones received only SGG for the same dose with an extention of duration until 30 days. 

>>>> Please add the meaning of "b.wt" in the abbreviation section 

3-

Ln 131 

For analyses of cytokines and antibodies, plasma of 20 blood donors was procured from the Ethiopian National Blood Bank. 

>>> PLEASE add : For control values of cytokines and antibodies analyses, plasma of 20 healthy blood donors was procured ....

4- 

In the method section from Ln 152

Serum/plasma samples were stored in a –20°C freezer when they were to be examined within 3-10 months; and/or in a –80°C deep freezer when they were to be analyzed after a longer period (i.e., >10 months). 

>>>> Is the ethical commitee agreed of this storage conditions ? 

5-

Moreover, english editing of the whole manuscript seems important to be done.

PLOS authors have the option to publish the peer review history of their article (what does this mean?). If published, this will include your full peer review and any attached files.

Reviewer #1: No

Reviewer #2: Yes: Issam Bennis

Figure Files:

Data Requirements:

Reproducibility:

References

---

## [Editor Report · Decision Letter 2]

7 Jul 2021

Dear Mr Tadesse,

We are pleased to inform you that your manuscript 'Antibody and cytokine levels in visceral leishmaniasis patients with varied parasitemia before, during and after treatment in patients admitted to Arba Minch General Hospital, southern Ethiopia' has been provisionally accepted for publication in PLOS Neglected Tropical Diseases.

Best regards,

Susan M Bueno

Associate Editor

Charles Jaffe

Deputy Editor

---

## [Editor Report · Acceptance letter]

30 Jul 2021

Dear Mr Tadesse,

We are delighted to inform you that your manuscript, "Antibody and cytokine levels in visceral leishmaniasis patients with varied parasitemia before, during and after treatment in patients admitted to Arba Minch General Hospital, southern Ethiopia," has been formally accepted for publication in PLOS Neglected Tropical Diseases.

Best regards,

Shaden Kamhawi

co-Editor-in-Chief

Paul Brindley

co-Editor-in-Chief
